# Circulating tumor DNA analysis depicts subclonal architecture and genomic evolution of small cell lung cancer

Jingying Nong[1], Yuhua Gong[2,3], Yanfang Guan[2,3], Xin Yi[2,3], Yuting Yi[2,3], Lianpeng Chang[2,3], Ling Yang[2,3], Jialin Lv[1], Zhirong Guo[4], Hongyan Jia[4], Yuxing Chu[2], Tao Liu[2,3], Ming Chen[5], Lauren Byers[6], Emily Roarty[6], Vincent K. Lam[6], Vassiliki A. Papadimitrakopoulou[6], Ignacio Wistuba[7], John V. Heymach[6], Bonnie Glisson[6], Zhongxing Liao[8], J. Jack Lee [9], P. Andrew Futreal[10], Shucai Zhang[1], Xuefeng Xia[11], Jianjun Zhang[6,10] & Jinghui Wang[1]

Subclonal architecture and genomic evolution of small-cell lung cancer (SCLC) under treatment has not been well studied primarily due to lack of tumor specimens, particularly longitudinal samples acquired during treatment. SCLC is characterized by early hematogenous spread, which makes circulating cell-free tumor DNA (ctDNA) sequencing a promising modality for genomic profiling. Here, we perform targeted deep sequencing of 430 cancer genes on pre-treatment tumor biopsies, as well as on plasma samples collected prior to and during treatment from 22 SCLC patients. Similar subclonal architecture is observed between pre-treatment ctDNA and paired tumor DNA. Mean variant allele frequency of clonal mutations from pre-treatment ctDNA is associated with progression-free survival and overall survival. Pre- and post-treatment ctDNA mutational analysis demonstrate that mutations of DNA repair and NOTCH signaling pathways are enriched in post-treatment samples. These data suggest that ctDNA sequencing is promising to delineate genomic landscape, subclonal architecture, and genomic evolution of SCLC.

[1] Department of Medical Oncology, Beijing Chest Hospital, Capital Medical University, Beijing Tuberculosis and Thoracic Tumor Research Institute, 101149 Beijing, China. [2] Geneplus-Beijing, 102206 Beijing, China. [3] Geneplus-Beijing Institute, 102206 Beijing, China. [4] Beijing Key Laboratory for Drug Resistance Tuberculosis Research, Beijing Chest Hospital, Capital Medical University, Beijing Tuberculosis and Thoracic Tumor Research Institute, 101149 Beijing, China. [5] Department of Radiation Oncology, Zhejiang Cancer Hospital, 310022 Hangzhou, China. [6] Department of Thoracic/Head and Neck Medical Oncology, University of Texas MD Anderson Cancer Center, Houston, TX 77030, USA. [7] Department of Translational Molecular Pathology, University of Texas MD Anderson Cancer Center, Houston, TX 77030, USA. [8] Department of Radiation Oncology, University of Texas MD Anderson Cancer Center, Houston, TX 77030, USA. [9] Department of Biostatistics, University of Texas MD Anderson Cancer Center, Houston, TX 77030, USA. [10] Department of Genomic Medicine, University of Texas MD Anderson Cancer Center, Houston, TX 77030, USA. [11] Houston Methodist Research Institute, Houston, TX 77030, USA. These authors contributed equally: Jingying Nong, Yuhua Gong, Yanfang Guan, Xin Yi. These authors jointly supervised this work: P. Andrew Futreal, Shucai Zhang, Xuefeng Xia, Jianjun Zhang, Jinghui Wang. Correspondence and requests for materials should be addressed to S.Z. (email: sczhang6304@163.com) or to J.Z. (email: jzhang20@mdanderson.org) or to J.W. (email: jinghuiwang2006@163.com)

Small-cell lung cancer (SCLC) accounts for ~15% of newly diagnosed lung cancers. SCLC is a very aggressive malignancy characterized by rapid growth and early hematogenous spread. At initial diagnosis, about 1/3 of patients presented with limited-stage disease (LD) that can be treated with chemoradiation, while the remaining patients presented with extensive-stage disease (ED), which are usually treated with palliative chemotherapy. Although an initial response to chemotherapy and/or radiotherapy can be achieved in most patients, nearly all patients recur with resistant disease, and the 5-year overall survival (OS) is 5–10%[1–4]. Clinical advances in SCLC remain an unmet need, as the treatment paradigm has not significantly changed over the past several decades[5].

Given the pattern of initial response to chemotherapy and/or radiotherapy and nearly invariable relapse, it has been speculated that treatment-naive SCLC harbors subclones of inherently refractory (resistant) cancer cells that give rise to the relapse in these patients. Therefore, delineating the subclonal architecture of SCLC and its molecular evolution during treatment by comparing genomic profiles of recurrent (and often chemo/radiation-resistant) SCLC tumors to paired treatment-naive tumors may provide new insight into the mechanisms underlying recurrence and therapeutic resistance and guide the development of novel treatment strategies.

Over the past decade, comprehensive genome-wide profiling has substantially advanced our understanding of the genomic landscapes of various cancer types and led to the identification of novel predicative/prognostic biomarkers and therapeutic targets[6–9]. However, compared to many other solid tumors, there have been only a few studies investigating the genomic landscape of SCLC[10–12]. This is primarily due to the lack of adequate tumor tissues because the majority of SCLC patients are not treated with surgical resection. Moreover, because recurrent SCLC usually progresses relatively quickly, recurrence suspected on imaging is typically followed by immediate second-line treatment without biopsy. Therefore, our knowledge of the genomic landscape of recurrent SCLC is very limited. There is an urgent need for alternative approaches for genomic profiling of SCLC, particularly in patients under treatment.

Sequencing circulating cell-free tumor DNA (ctDNA)—fragmented DNA shed from tumors into circulating system—may be such an alternative. ctDNA sequencing was reported in many cancer types to have potentials in disease monitoring[13–15] and detection of minimal residual disease[16,17]. Because SCLC has a rapid growth rate and is highly metastatic with early hematogenous spread, ctDNA may be readily detectable in SCLC patients. In addition, since ctDNA sequencing is noninvasive and "real-time", it could be an ideal tool for investigating genomic evolution of SCLC over time, particularly during treatment. Herein, we report a study on deep sequencing of 430 cancer-related genes in 43 ctDNA samples collected prior to treatment and at different time points during treatment from 22 SCLC patients. We show that circulating tumor DNA sequencing is promising to delineate genomic landscape, subclonal architecture and investigate genomic evolution of small-cell lung cancer under therapy.

## Results

### Genomic profiling of SCLC from pre-treatment ctDNA. To investigate whether it is feasible to perform genomic profiling of ctDNA from SCLC patients, pre-treatment plasma samples from 22 SCLC patients (Supplementary Data 1) were subjected to DNA extraction and next-generationsequencing (NGS) of all coding exons and selected introns of 430 cancer genes (Supplementary Data 2) with a target region about 2.3 Mb for an average

sequencing depth of 873× (538×–1169×). DNA from paired peripheral blood mononuclear cells of the same patient was sequenced as the germline control. A total of 342 somatic mutations were identified with a median of 16 mutations per sample (ranging from 5 to 38) and an average mutation burden of 6.8 per Mb (Supplementary Data 3), which are comparable to previous reports[12]. C > A transversions that are associated with smoking and C > T transitions that may be associated with aging[18] were the dominant aberrations, accounting for 39.5% and 27.2% of the total somatic SNVs, respectively. Twenty-nine genes were found to be mutated in more than 10% of patients (Fig. 1), including many commonly mutated genes in SCLC[12]. As expected, TP53 and RB1 are the most frequently mutated genes in this cohort of patients. TP53 gene point mutations were identified in 91% of patients (20/22), including 4 patients with mutations in both alleles based on informative SNPs (Supplementary Fig. 1A). Point mutations in RB1 were identified in 64% (14/22) of patients (Supplementary Fig. 1A). In addition, loss of heterozygosity (LOH) of TP53 was detected in one patient and LOH of RB1 was found in five patients based on informative SNPs (Supplementary Fig. 1A). Mutations in other frequently mutated genes in SCLC such as NOTCH1–4, CREBBP, and EP300 (Supplementary Fig. 1B) and copy number alterations of MYC, MYCL1, and MYCN[12] were also observed in our cohort (Supplementary Fig. 1C).

### Concordance of somatic mutations between tumor DNA and ctDNA.
Although encouraging progress has been made, reliability of ctDNA sequencing is still in question and tumor tissue sequencing remains the gold standard. In a previous report, the concordance rate between tumor DNA and ctDNA was as low as 12%[19] highlighting the technical challenges in ctDNA sequencing technologies. To assess the reliability of the ctDNA assay applied in this study, genomic profiling using the same gene panel and same sequencing platform was performed with DNA of paired pre-treatment tumor samples from eight patients, where tissues were available, for an average sequencing depth of 870× (348×–1281×). Somatic mutations were identified in all eight tumors with an average of 13 mutations per sample (ranging from 3 to 26, Supplementary Data 4). Overall, a median of 94% of mutations (ranging from 0 to 100%) detected in tumor DNA were also detected in paired ctDNA samples (Fig. 2 and Supplementary Fig. 2) suggesting ctDNA sequencing is sensitive for detecting somatic mutations in the majority of SCLC patients in this cohort. Patient CA170 was an exception in that none of the 26 mutations detected in tumor DNA were detected in matched plasma DNA at the initial sequencing depth of 943×. Two of the 26 mutations were detected by increasing the sequencing depth to 1866× along with 17 other mutations that were present in ctDNA but below the calling threshold.

Furthermore, the variant allelic frequencies (VAF) of the 69 shared mutations between ctDNA and paired tumor DNA moderately correlated to each other (Spearman $r = 0.558$, $p < 0.0001$) (Supplementary Fig. 3) indicating that the genomic landscape derived from ctDNA reflects that from SCLC tumors to a certain degree. However, a subset of mutations was exclusively detected in ctDNA (Fig. 2), with a median concordance rate of 60% (ranging from 5 to 77%). Taken together, these data suggested that there is genomic intra-tumor heterogeneity (ITH) of SCLC in this cohort of patients and highlighted the potential advantage of sequencing ctDNA over a single tumor biopsy to reveal the global genomic landscape of SCLC.

### Subclonal architecture of ctDNA from SCLC.
The variable VAFs in different mutations from ctDNA implied variable

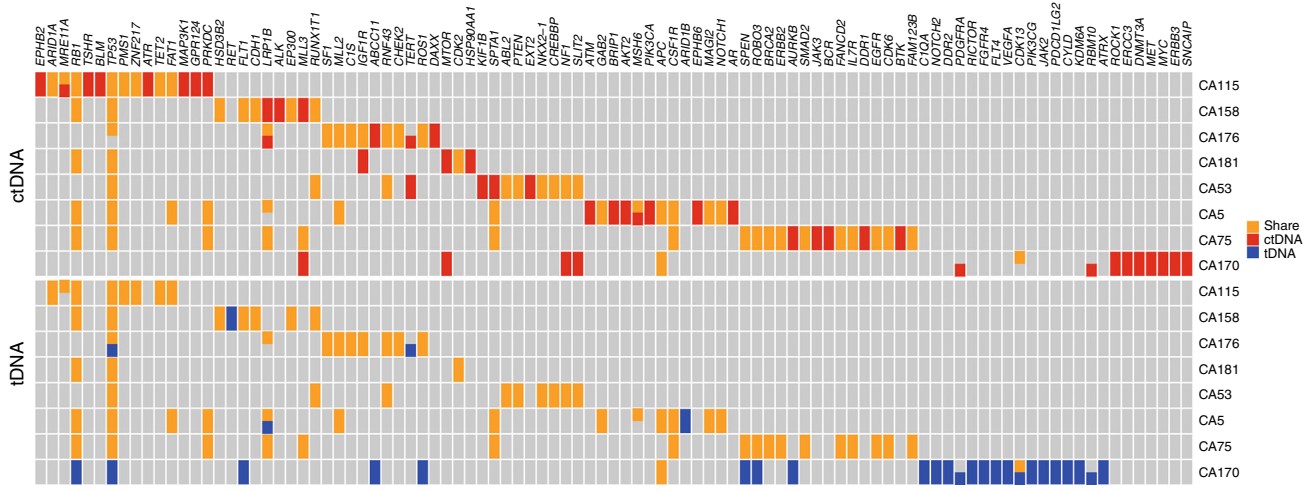

**Fig. 1** Somatic mutation profiles of 22 SCLC patients from pre-treatment ctDNA sequencing of 430 cancer genes. Twenty-two patients were arranged along the x-axis. Mutation per Mb region, clinical and pathological characters were shown in the upper panel. Genes with somatic mutations were shown in the middle panel. Mutation frequencies of each gene were shown on the left and mutation frequencies of these genes in previous report[12] were shown on the right to each gene. The mutational spectrum was shown at the bottom

**Fig. 2** Somatic mutations detected in paired tumor DNA and ctDNA. Genes with somatic mutations were listed on the x-axis, and samples were shown on the y-axis. Mutations detected only in tumor DNA (tDNA), only in ctDNA or in both were shown in blue, red and orange, respectively

clonality of different mutations. To explore the subclonal architecture of SCLC from ctDNA sequencing, we used PyClone[20] to infer cancer cell fraction (CCF) of each mutation in each ctDNA sample. Mutations were then clustered based on corresponding CCF, and subclonal architecture of ctDNA from SCLC was

subsequently inferred[20]. The results demonstrated distinct subclonal architecture in different patients. Different numbers of mutation clusters were present in individual patients with a median of 11 (ranging from 2 to 26) clusters per patient. Of particular interest, the subclonal architecture derived from

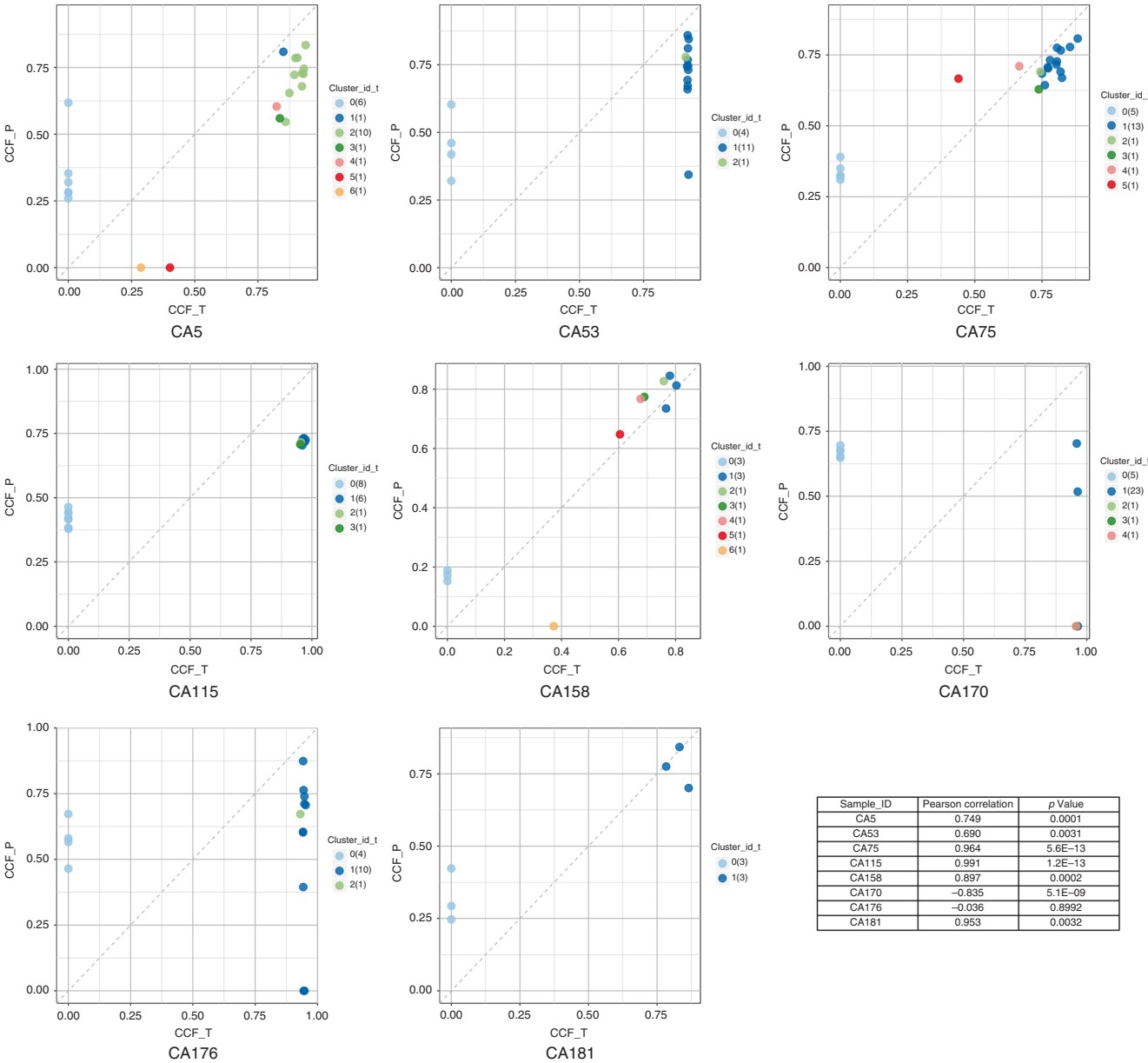

**Fig. 3** Comparison of genomic architecture derived from paired ctDNA versus tumor DNA. CCF of mutations were calculated in ctDNA and tumor DNA. Each dot represents one mutation and the color of each dot indicates the subclone that given mutation was clustered to. Correlation of CCF of all mutations in each pair of samples was shown in the table at lower-right corner. CCF_P: CCF of mutations in ctDNA; CCF_T: CCF of mutations in tumor DNA

ctDNA demonstrated high similarity to those derived from paired tumor DNA in 6 of the 8 patients (Fig. 3) and the number of mutation clusters was correlated between tumor DNA and ctDNA (Spearman $r = 0.7099$, $p = 0.0485$). These results suggest that ctDNA may be a reasonable alternative to tumor DNA for delineating the subclonal architecture of SCLC. On the other hand, ctDNA demonstrated more mutation clusters than tumor DNA (median of 11 clusters per ctDNA sample versus 4 per tumor DNA sample, $p = 0.047$, wilcoxon matched pairs signed rank test) highlighting again the advantage of ctDNA over single biopsies in revealing the global genomic landscape of SCLC.

**Association between ctDNA and clinical parameters.** Next, we assessed whether ctDNA is associated with patient characteristics. We did not find any association between the yield of DNA per 1 ml plasma sample, total mutation burden, mutations of particular genes, or VAF of certain mutations with patient age, gender,

smoking status, cancer stage, recurrence status, or survival, although the small sample size limits the power to detect these associations. In view of the substantial variation in ctDNA subclonal architecture as mentioned above, we next investigated whether ctDNA subclonal architecture correlated with clinicopathological parameters. We utilized the average VAF of mutations from the major clones (the cluster with the greatest CCF) as a surrogate for overall ctDNA level. Varied ctDNA levels were observed among patients, with a median value of 0.18 (95% CI, 0.08 to 0.36) and ctDNA level was moderately correlated with tumor burden (sum of the longest diameters of the target lesions on CT scans) (Spearman $r = 0.413$, $p = 0.056$, Supplementary Fig. 4). Of particular interest, patients with higher than median ($\geq 0.18$) ctDNA level had significantly shorter progression-free survival (PFS) and OS ($p = 0.002$ for PFS and $p = 0.012$ for OS; Fig. 4). The median PFS among patients with higher versus lower than median ctDNA level was 5.3 months (95% CI,

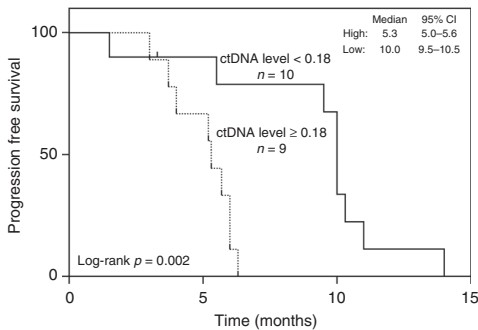
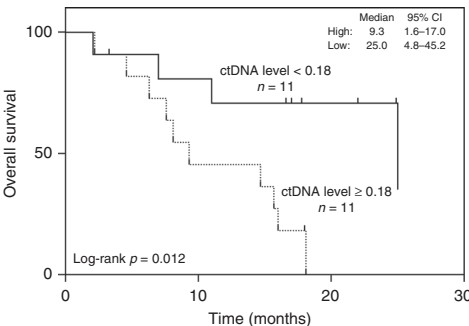

**Fig. 4** The association of PFS and OS with ctDNA level measured by the average VAF of clonal mutations. Left, Patients were divided into two groups by ctDNA level. High ctDNA level group (higher than median of 0.18, dot line) was significantly associated with shorter PFS. Three patients were excluded from PFS analysis for three different reasons, including one patient who did not receive any treatment based on the patient's choice, one patient received surgery, and one could not offer out-patient otherapeutic records. Right, Patients were divided into two groups using the median ctDNA level of 0.18. Significantly shorter OS was observed in patients with higher ctDNA level (dot line). Small-cell lung cancer (SCLC) may evolve under treatment. But tumor tissues are often not available to study evolution of SCLC. Here, the authors utilize circulating tumor DNA to investigate the genomic evolution and subclonal architecture of SCLC during therapy

5.0–5.6 months) versus 10.0 months (95% CI, 9.5–10.5 months) (Fig. 4, left panel). The median OS was over 15 months shorter in the high (9.3 months; 95% CI, 1.6–17.0 months) versus low (25.0 months; 95% CI, 4.8–45.2 months) ctDNA level group (Fig. 4, right panel). The difference remains significant in multivariate analysis after adjusting for age (<60 or ≥60), stage (LD or ED) and tumor burden (HR = 8.4, $p = 0.008$ for PFS and HR = 4.7, $p = 0.021$ for OS; Supplementary Data 5). In our cohort, there are three patients with ED disease who remain alive for longer than 20 months. ctDNA levels of these three patients are all below 0.18 (0.04, 0.08, and 0.12, respectively). In addition, comparing samples collected at different time points during the clinical course, we observed dynamic changes in ctDNA levels, which are well correlated with tumor measurements on imaging (Supplementary Fig. 5), suggesting that ctDNA sequencing has the potential for monitoring the clinical course of SCLC.

**Depicting genomic evolution of SCLC under therapy using ctDNA.** Although SCLC is very sensitive to initial therapy, nearly all patients experience relapse with broadly resistant cancer. Thus, investigating the molecular evolution of SCLC during treatment may provide new insight into the mechanisms underlying therapeutic resistance and guide the development of novel treatment strategies. To that end, we compared the genomic profiles derived from pre-treatment ctDNA to the genomic profiles from ctDNA at different time points during treatment from 11 patients with post-treatment plasma samples available. As shown in Supplementary Data 6 and Supplementary Fig. 6, variable dynamic changes were observed in different mutation clusters from each individual patient. In general, the average VAF of major mutation clusters substantially decreased after platinum treatment, possibly reflecting the decrease in tumor burden with treatment in these patients. However, certain clones subsequently reemerged (e.g., in patients CA5, CA44, and CA141) coinciding with disease progression. Mutations were detected in 9 of 11 patients with post-treatment samples. Thirty-three mutations were exclusively detected in the post-treatment samples and 13 mutations clustered into subclones that were dominant in the post-treatment compared to pre-treatment samples (Supplementary Data 6). Among mutations exclusively detected in post-treatment samples, there were 3 truncating mutations and 17 missense mutations predicated to impact the functions of associated genes by PolyPhen-2[21] and 9 genes were previously reported to be associated with chemoresistance in several types of cancer (Supplementary Data 7).

## Discussion

Mutations from ctDNA are usually easier to detect in late-stage malignancies than early-stage diseases[13]. However, SCLC is a rapidly proliferating malignancy with early hematogenous spread[5], suggesting that ctDNA might be readily detectable in SCLC patients, regardless of stage. Indeed, we detected mutations in all pre-treatment plasma samples in this cohort of patients at a relatively low sequencing depth compared to previous studies on ctDNA sequencing of solid tumors[22,23]. A median of 94% of mutations detected in paired tumor samples was also detected in the plasma samples in our cohort. However, only 5 mutations were detected from the plasma of patient CA170 and none of the 26 mutations detected in the tumor sample was also detected in the plasma sample at the initial sequencing depth. At a higher sequencing depth, only 2 of the 26 mutations in tumor were also detected in plasma despite the total mutations detected in plasma increasing to 15. One possible explanation is that the majority of mutations detected in tumor and plasma were subclonal and restricted to spatially separated tumor regions. Furthermore, CA170 had a 2.3 cm primary tumor without lymph node or distant metastasis (AJCC stage IA: T1bN0M0) and is alive 17 months post resection without evidence of recurrence. In comparison, CA181, another patient with limited-stage disease (AJCC stage IIIA), had only three mutations detected in the tumor, all of which were also detected in the corresponding plasma sample. These results suggest that the sensitivity of ctDNA sequencing depends on the proportion of tumor-derived DNA in the plasma, which is influenced by factors such as tumor burden[24] and subclonal architecture.

Subclonal architecture may have profound impact on cancer biology and thus clinical outcomes of cancer patients. Recent studies[25–27] have reported on the potential clinical and biological impact of the genomic subclonal architecture of non-small-cell lung cancer. The genomic subclonal architecture of SCLC has not been well studied, primarily due to the lack of large resected tumor specimens for multiregion sequencing. In this study, we demonstrated very similar subclonal architecture between ctDNA and paired tumor in most patients, suggesting that ctDNA can be used not only to detect somatic mutations, but also to study subclonal architecture of SCLC. In addition, ctDNA levels as measured by the average VAF of clonal mutations, rather than

any single gene mutation, were found to be associated with PFS and OS, suggesting that subclonal architecture may provide deeper prognostic information than any single gene mutation.

Shedding of ctDNA is a complicated process affected by many factors and tumor burden is one of the most important determinants[24]. On the other hand, tumor burden is also a well-known prognostic factor associated with patient survival across different cancer types[28–31]. Therefore, it is not unreasonable to hypothesize that ctDNA levels may simply reflect tumor burden that in turn is associated with survival. We did observe a moderate correlation (Spearman $r = 0.413$, $p = 0.056$, Supplementary Fig. 4) between pre-treatment ctDNA levels and tumor burden. However, we did not observe an association between tumor burden and survival ($p = 0.184$ for PFS and $p = 0.367$ for OS, Supplementary Data 5). It is likely due to the small sample size of this cohort. Because of the limitation of current imaging modality, particularly in detecting small metastases, tumor burden measurement is not optimal. On the other hand, patient survival can be impacted by many factors including tumor burden, patient age, and treatment etc. Therefore, a large cohort will be needed to reveal the correlation between tumor burden and survival. Furthermore, tumor stage is one of the most important prognostic factors for SCLC. In the current study, the association between ctDNA levels measured by average VAF of clonal mutations and survival held true in multivariate analyses after adjusting for major prognostic factors including stage, age, and tumor burden (Supplementary Data 5), suggesting that genomic subclonal architecture may provide important prognostic information independent of these factors.

The current standard of care for therapeutic response assessment and disease monitoring for SCLC patients is cross-sectional imaging. While it is an essential clinical tool, imaging has important limitations. For examples, it has low sensitivity/specificity when tumor size is <10 mm; it can be problematic to distinguish disease progression from treatment effects, infection, or inflammatory changes; it does not reflect treatment-induced changes in tumor genotype. Therefore, real-time and highly accurate modalities for therapeutic response assessment and monitoring of tumor burden are needed to provide early insight into treatment efficacy and recurrence of disease. ctDNA sequencing has shown its potential in detection and disease monitoring in multiple cancer types[15,17,24,32]. In this study, we have demonstrated that ctDNA level correlates well with changes in tumor on computerized tomographic imaging (Supplementary Fig. 5). Given the early hematogenous spread of SCLC and rapid advancement in sensitivity, as well as specificity of ctDNA sequencing technologies, ctDNA analysis also has the potential to serve as a real-time and highly sensitive modality for SCLC monitoring. Studies are ongoing to assess whether ctDNA can detect disease recurrence/progression prior to conventional imaging studies and whether early detection will improve clinical outcomes.

The biology behind the broadly resistant phenotype of recurrent SCLC is well-documented, but poorly understood. Molecular aberrations that are specific to, or enriched (i.e., change from subclonal to clonal) in, recurrent tumors compared to treatment-naive tumors are potential candidates for driving this broad therapeutic resistance. Due to the difficulties inherent in sampling tumor at progression, serial analysis of plasma DNA before and after treatment provides an opportunity to study the genomic evolution of SCLC during and after treatment. Using this approach, we identified many mutations that were enriched in the post-treatment samples and some of these genes, such as *NOTCH1*, *ERCC1*, and *STED2*, have been previously reported to be associated with chemoresistance[33–36]. As many of the post-treatment plasma samples in this study were collected long before clinical recurrence, we cannot conclude that these mutations are associated with therapeutic resistance. In addition, the enrichment of mutations in post-treatment samples could also be confounded by small sample size and mutagenic effects from chemotherapy and/or radiation, therefore, comparing genomic landscape of longitudinally collected SCLC samples before and after treatment is needed to validate these intriguing findings. Nevertheless, our results suggested that ctDNA analysis for investigating the genomic evolution of SCLC is feasible.

A significant obstacle to advancing translational SCLC research is the challenge in obtaining tumor material. In this study, we provide proof-of-concept evidence that ctDNA sequencing may be a reliable modality for detecting somatic mutations, depicting the subclonal architecture and investigating the genomic evolution of SCLC. To our knowledge, this is the first study that employed ctDNA sequencing in SCLC patients using a large gene panel capable of identifying sufficient mutations for subclonal analysis. Another foreseen advantage of a large gene panel is a more accurate estimation of tumor mutation burden (TMB), a potential biomarker for response to immune checkpoint blockade in many cancer types[37–41]. Recently presented early phase clinical trials have shown encouraging activity of immune checkpoint blockade in treatment of SCLC patients[42–44], and high TMB has been demonstrated to be associated with superior response and prolonged survival in SCLC[45]. Therefore, ctDNA sequencing using a large gene panel, such as the one presented here, could be used to assess TMB for SCLC patients, particularly when biopsy is not clinically feasible or the quantity and/or quality of biopsy is not satisfactory.

## Methods

**Patients and sample collection.** Patients with SCLC diagnosed at Beijing Chest Hospital between 2014 and 2015 were enrolled. The study was approved by the institutional review board and all patients provided written informed consents. A volume of 10 ml of blood was collected within 2 weeks before treatment from each patient. Serial post-treatment blood sample collection was also planned before the 2nd, 3rd, 4th, and 5th cycle of chemotherapy and at disease progression for each patient. Tumor burden were calculated according to RECIST 1.1 (sum of the longest diameters of the target lesions on CT scans). In addition, pre-treatment formalin-fixed paraffin-embedded (FFPE) specimens obtained by biopsy or surgical resection were collected from eight patients with a median of 5-day interval (0 day to 10 days) between the collection of pre-treatment blood samples.

**Sample processing and DNA extraction.** Peripheral blood was collected in EDTA Vacutainer tubes (BD Diagnostics, Franklin Lakes, NJ, USA) and processed within 4 h. Plasma was separated by centrifugation at 1600× *g* for 10 min, transferred to new microcentrifuge tubes, and centrifuged at 16,000× *g* for 10 min to remove remaining cell debris. Peripheral blood lymphocytes (PBLs) from the first centrifugation were used for the extraction of germline genomic DNA. PBL DNA was extracted using the DNeasy Blood & Tissue Kit (Qiagen, Hilden, Germany). DNA was isolated from plasma using QIAamp Circulating Nucleic Acid Kit (Qiagen, Hilden, Germany). Genomic DNA was extracted from FFPE samples using Maxwell® RSC DNA FFPE Kit (Promega, Madison, WI, USA). DNA concentration was measured using a Qubit fluorometer and the Qubit dsDNA HS (High Sensitivity) Assay Kit (Invitrogen, Carlsbad, CA, USA). The size distribution of plasma DNA was assessed using an Agilent 2100 BioAnalyzer and the DNA HS kit (Agilent Technologies, Santa Clara, CA, USA).

**Sequencing library construction and target enrichment.** Before library construction, 1 µg each of genomic DNA extracted from PBL and FFPE specimen was sheared to 300-bp fragments with a Covaris S2 ultrasonicator (Covaris, Woburn, MA, USA). A volume of 20–80 ng DNA from plasma were used for library construction. Indexed Illumina next-generation sequencing (NGS) libraries were prepared from PBL DNA, tumor DNA, and plasma DNA using the KAPA Library Preparation Kit (Kapa Biosystems, Wilmington, MA, USA).

Target enrichment was performed with a custom SeqCap EZ Library (Roche NimbleGen, Madison, WI, USA). To explore the comprehensive genetic properties of SCLC, the capture probe was designed based on ~2.3 Mb genomic regions of 430 genes frequently mutated in SCLC and other common solid tumors. Capture hybridization was carried out according to the manufacturer's protocol. Following hybrid selection, the captured DNA fragments were amplified and then pooled to generate several multiplex libraries.

**NGS sequencing**. Sequencing was carried out using Illumina $2 \times 75$ bp paired-end reads on an Illumina HiSeq 3000 instrument according to the manufacturer's recommendations using TruSeq PE Cluster Generation Kit v3 and the TruSeq SBS Kit v3 (Illumina, San Diego, CA, USA).

**Sequence data analysis**. After removal of terminal adaptor sequences and low-quality data, reads were mapped to the reference human genome (hg19) and aligned using BWA (0.7.12-r1039)[46]. MuTect2 (3.4–46-gbc02625)[47] was employed to call somatic small insertions and deletions (InDels) and single nucleotide variants (SNVs). Contra (2.0.8) was used to detect copy number variations[48] and an algorithm was used to identify LOH based on informative SNPs[49]. For structure variations (SV), baits were designed to capture selected exons and introns of *RET*, *ALK*, *ROS1*, and *NTRK1* oncogenes based on previously reported SVs in these genes and an in-house algorithm was used to identified split-read and discordant read-pair to identify SVs. All final candidate variants were manually verified with the integrative genomics viewer browser.

**Subclonal analysis**. Pyclone was employed to infer the subclonal architecture of all DNA samples from plasma and available SCLC tumors[20]. The copy number information of each SNV was used as input for PyClone analysis[27,50] and the CCF was inferred and variants were clustered as previously described[20]. PyClone was run with 20,000 iterations and default parameters. For comparing CCF in matched tumor DNA and ctDNA, the parental_copy_number mode and a burn-in of 2000 were added. Variants located in the cluster with greatest mean CCF were defined as clonal, the rest were subclonal[20].

**Statistical analysis**. Survival analysis was performed by multivariate Cox proportional hazards regression analysis and Kaplan–Meier survival analysis with log-rank test. IBM SPSS software (23.0) and GraphPad Prism (6.01) were used in statistical analysis. All tests were two-sided and considered statistically significant at $p < 0.05$.

**Data availability**. All mutations reported in this study were provided in the supplementary information. De-identified patient clinical information was provided in the supplementary information. All other relevant data could be obtained from the corresponding authors of this study.

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

## Acknowledgements

This study was supported by MD Anderson Lung cancer Moon Shot Program, MD Anderson Physician Scientist Program, Khalifa Scholar Award, the Cancer Prevention and Research Institute of Texas (R120501), the Cancer Prevention and Research Institute of Texas Multi-Investigator Research Award grant (RP160668), T.J. Martell Foundation, NIH CCSG (P30 CA016672), Sabin Family Foundation Award. Dr. Tao Xu kindly provided advice on Cox proportional hazards regression analysis.

## Author contributions

J.H.W., J.Z., X.F.X., S.C.Z., and P.A.F. designed the study. J.Y.N, J.L.L., Z.R.G., H.Y. J., and J.H.W. enrolled the patients and collected all the specimens as well as the patients clinical informations, they also followed up all participant. Y.H.G., Y.F.G., X.Y., Y.T.Y., L. Y., and Y.X.C. performed the sequencing and data analysis. Y.H.G., L.P.C., T.L., J.Z., X.Y., J.Y.N., M.C., and J.H.W. interpreted results. J.J.L. supervised all statistical analyses. Y.H. G., J.Z., J.Y.N., L.B., E.R., V.K.L., V.A.P., I.W., J.V.H., B.G., Z.X.L., and P.A.F. wrote the manuscript. All authors read and approved the final manuscript.

## Additional information

**Competing interests:** The authors declare the following competing interests: Y.H.G., Y.F. G., X.Y., Y.T.Y., L.P.C., L.Y., Y.X.C., and T.L. are current employees of Geneplus-Beijing. X.Y. and L.Y. hold leadership positions and stocks of Geneplus-Beijing. J.V.H. is a consultant for AstraZeneca, Abbvie, Boehringer Ingelheim, Bristol-Myers Squibb, Medivation, ARIAD, Synta, Oncomed, Novartis, Genentech, and Calithera Biosciences, holds stock in Cardinal Spine LLC and Bio-Tree, and has received funding from AstraZeneca. J.Z. is a consultant for AstraZeneca and receives honoraria from Bristol-Myers Squibb. I.I.W. receives honoraria from Roche/Genentech, Ventana, GlaxoSmithKline, Celgene, Bristol-Myers Squibb, Synta Pharmaceuticals, Boehringer Ingelheim, Medscape, Clovis, AstraZeneca, and Pfizer, and research support from Roche/ Genentech, Oncoplex, and HGT. The remaining authors declare no competing interests.

