## [Peer Review File · Nature Communications]

Reviewers' comments:

Reviewer #1 (Remarks to the Author):

The manuscript describes ctDNA Next Generation Sequencing (NGS) analysis of 22 SCLC patients encompassing 43 ctDNA samples and using a targeted NGS panel directed towards 430 cancer-related genes. Tumor specific changes were picked up for all patients and the observations reported are both novel and of broad scientific/clinical interest well deserving of publication.

One of the key findings was that mean variant allele frequency (VAF) of clonal mutations in pre-treatment ctDNA was associated with progression-free survival (PFS) and overall survival (OS). In addition they also show that ctDNA levels are also correlated with tumor measurements obtained by computerized tomographic imaging. It would be useful if the authors could further clarify or comment on the relationship between ctDNA VAF, tumor measurements, PFS and OS – are ctDNA levels simply reflecting overall tumor burden that in turn is associated with PFS and OS?

The authors also state that mutations of DNA repair and NOTCH signaling are enriched in post-treatment samples and that this may reflect the selective pressure of chemotherapy and/or radiation. However, given the small number of patient samples involved and the fact that treatment itself is mutagenic it is difficult to argue for this and would recommend a more cautious phrasing.

Reviewer #2 (Remarks to the Author):

In the study "Circulating tumor DNA analysis depicts sub clonal architecture and genomic evolution of small cell lung cancer" Nong et al. performed targeted exome sequencing (430 gene panel) on 43 ctDNA samples from 22 patients with small cell lung cancer both prior to treatment and during treatment in subset of patients. They performed similar analysis on pretreatment tumor samples from 8 patients. The authors identified the frequency of genomic alterations in these samples as well as assessed the clonal architecture of these alterations. In a subset of 11 patients they compared pre-treatment ctDNA analyses to samples taken during treatment. The authors found a correlation with average VAF of clonal alterations with PFS and OS. Overall this represents a thorough study of a relatively understudied cancer, and is therefore of particular interest to the field. The findings could be strengthened by addressing the following concerns:

1. Paired tumor specimens were only available for 8 of the 22 patients analyzed. This makes interpreting comparisons between ctDNA and tumor DNA difficult. The addition of more tumor specimens to the analysis would strengthen the findings. The authors should calculate sensitivity and specificity of ctDNA compared to tumor DNA to identify genomic alterations.
2. The authors perform an analysis of clonal vs. sub clonal alterations using Pyclone. Pyclone relies on comprehensive copy-number analysis to define clonal architecture. The authors use a 430 gene panel which does not provide a thorough copy number analysis of the genome. Please explain or provide data showing that the copy-number analysis of 430 genes is adequate for Pyclone analysis. A supplemental figure defining which specific genomic alterations are clonal vs. sub clonal would better illustrate the findings.
3. The correlation with the average VAF of clonal alterations with PFS and OS is intriguing. However, this could be explained by differences in overall tumor burden between patients with better or worse outcomes. Other than controlling for limited vs. extensive stage, this should be controlled for. Also a correction for multiple hypothesis testing should be performed as the authors assessed multiple variables for clinical outcomes correlation and this was the only one that was positive.

4. The authors identify 33 mutations that were seen only in post-treatment samples. The authors should more clearly define which of these are novel and not previously implicated in small cell lung cancer chemo resistance. The study would be strengthened by functional assays assessing whether these alterations contribute to chemo resistance.

Referees' comments:**Reviewer #1 (Remarks to the Author):**

The manuscript describes ctDNA Next Generation Sequencing (NGS) analysis of 22 SCLC patients encompassing 43 ctDNA samples and using a targeted NGS panel directed towards 430 cancer-related genes. Tumor specific changes were picked up for all patients and the observations reported are both novel and of broad scientific/clinical interest well deserving of publication.

Author's response: We appreciate reviewer's favorable comments.

1. One of the key findings was that mean variant allele frequency (VAF) of clonal mutations in pre-treatment ctDNA was associated with progression-free survival (PFS) and overall survival (OS). In addition they also show that ctDNA levels are also correlated with tumor measurements obtained by computerized tomographic imaging. It would be useful if the authors could further clarify or comment on the relationship between ctDNA VAF, tumor measurements, PFS and OS – are ctDNA levels simply reflecting overall tumor burden that in turn is associated with PFS and OS?

Author's response: We appreciate this excellent comment and we thank the reviewer for bringing up this very important point that is worth further discussion. We agree with the reviewer that it is very reasonable to hypothesize that 1) tumor burden is associated with shedding of ctDNA; and 2) tumor burden is associated with patient survival. Indeed, we did observe a moderate correlation (Spearman $r=0.413$; $p=0.056$) between pretreatment ctDNA levels and tumor burden (measured by the sum of the longest diameters of the target lesions on CT scans according to RECIST 1.1). However, we did not observe an association between tumor burden and PFS ($p=0.184$) or OS ($p=0.367$) in our cohort (Supplementary Table 5). It is likely due to the small sample size of this study. Because of the limitation of current imaging modality, particularly in detecting small metastases, tumor burden measurement is not optimal. On the other hand, patient survival can be impacted by many factors such as overall disease burden, patient age, treatment etc. Therefore, a large cohort will be needed to reveal the correlation between tumor burden and survival. We have added a supplementary figure (Supplementary Fig. 4), updated Supplementary Table 5, results (page 10, line 180-182, 190-191) and methods (page 18, line 339-340), and added discussion (page 14, line 258-276) about this point in the revised manuscript.

2. The authors also state that mutations of DNA repair and NOTCH signaling are enriched in post-treatment samples and that this may reflect the selective pressure of chemotherapy and/or radiation. However, given the small number of patient samples involved and the fact that treatment itself is mutagenic it is difficult to argue for this and would recommend a more cautious phrasing.

Author's response: We appreciate this comment and we agree with the reviewer that the enrichment of mutations in post-treatment samples could be confounded by small sample size and mutagenic effect from chemotherapy and/or radiation. We have updated discussion (page 16, line 308-312) regarding to this accordingly in the revised manuscript.

Reviewer #2

In the study "Circulating tumor DNA analysis depicts sub clonal architecture and genomic evolution of small cell lung cancer" Nong et al. performed targeted exome sequencing (430 gene panel) on 43 ctDNA samples from 22 patients with small cell lung cancer both prior to treatment and during treatment in subset of patients. They performed similar analysis on pretreatment tumor samples from 8 patients. The authors identified the frequency of genomic alterations in these samples as well as assessed the clonal architecture of these alterations. In a subset of 11 patients they compared pre-treatment ctDNA analyses to samples taken during treatment. The authors found a correlation with average VAF of clonal alterations with PFS and OS. Overall this represents a thorough study of a relatively understudied cancer, and is therefore of particular interest to the field. The findings could be strengthened by addressing the following concerns:

Author's response: We appreciate reviewer's favorable comments.

1. Paired tumor specimens were only available for 8 of the 22 patients analyzed. This makes interpreting comparisons between ctDNA and tumor DNA difficult. The addition of more tumor specimens to the analysis would strengthen the findings. The authors should calculate sensitivity and specificity of ctDNA compared to tumor DNA to identify genomic alterations.

Author's response: We totally agree with the reviewer on the limitation that only 8 of 22 patients had paired tumor available. We were planning to compare ctDNA and paired tumors from all patients. However, only 8 patients had sufficient remaining tissues for next generation sequencing after histologic diagnosis. Inadequate tissue is unfortunately a common problem for small cell lung cancer highlighting the urgent need for an alternative approach for genomic profiling such as ctDNA sequencing. Nevertheless, we calculated the sensitivity of ctDNA analysis using tumor DNA as gold standard and found a median sensitivity of 94% (page 8, line 131-132) as illustrated by Venn diagrams below (Supplementary Fig. 2). We did not report specificity in the manuscript because of the following two reasons: 1) As the vast majority of nucleotides are wide-type in both ctDNA and tumor tissues (true negative), while the number of potential false positive (wild-type in tumor tissues and mutated in ctDNA samples) was relatively small, the specificity (Specificity = true negative/(true negative + false positive)×100%) will be very high for any assay with a relatively large gene panel. The specificity was over 99% in all 8 pairs in our study. 2) When a mutation is detected in ctDNA, but not in paired tumor tissue, it is not necessarily false positive as it can be due to intra-tumor heterogeneity.

2. The authors perform an analysis of clonal vs. sub clonal alterations using Pyclone. Pyclone relies on comprehensive copy-number analysis to define clonal architecture. The authors use a 430 gene panel which does not provide a thorough copy number analysis of the genome. Please explain or provide data showing that the copy-number analysis of 430 genes is adequate for Pyclone analysis. A supplemental figure defining which specific genomic alterations are clonal vs. sub clonal would better illustrate the findings.

Author's response: We thank the reviewer for bringing up this very important technical point. In our study, the copy number information of each SNV was used as input for PyClone analysis as previously described (Murtaza, M., et al., Multifocal clonal evolution characterized using circulating tumour DNA in a case of metastatic breast cancer. *Nat. Commun.*, 2015; Jamal-Hanjani M., et al., Tracking the evolution of non-small-cell lung cancer. *N Engl J Med.* 2017). Except of CNV mentioned in the manuscript (page 7, line 118 and Supplementary Fig. 1C), other copy neutral region was considered as 2 copies (diploid). We defined those mutations located in the cluster with greatest mean cellular prevalence as clonal mutations that may have been acquired during early carcinogenesis, while the subclonal mutations that may have been acquired later in molecular time with smaller cellular prevalence (Roth et al., PyClone: statistical inference of clonal population structure in cancer, *Nature Methods.* Supplementary Figure 1, see below). We accordingly clarified the definition of clonal mutation in the methods (page 21, line 396-397, 400-402) and referred to this. In addition, we updated Supplementary Table 6 and added a column to mark each mutation as clonal or subclonal.

Supplementary Figure 1: Clonal evolution model | (a) A hypothetical phylogenetic tree generated by clonal expansion via the accumulation of mutations (stars). Unlike traditional phylogenetic trees internal nodes (clones) in the tree may contribute to the observed data, not just the leaf nodes. (b) Hypothetical observed cellular prevalences for the mutations in tree. Mutations occurring higher up the tree always have a greater cellular prevalence than their descendants (the same statement need not be true about variant allelic prevalence because of the effect of genotype). Note that the green, blue and purple mutations occur at the same cellular prevalence because they always co-occur in the clones of the tree.

3. The correlation with the average VAF of clonal alterations with PFS and OS is intriguing. However, this could be explained by differences in overall tumor burden between patients with better or worse outcomes. Other than controlling for limited vs. extensive stage, this should be controlled for. Also a correction for multiple hypothesis testing should be performed as the authors assessed multiple variables for clinical outcomes correlation and this was the only one that was positive.

Author's response: We thank the reviewer for pointing out this critical issue. We agree with the reviewer that it is very reasonable to hypothesize that larger tumor burden is associated with inferior survival. However, we did not observe an association between tumor burden and PFS ($p=0.184$) or OS ($p=0.367$) in our cohort (Supplementary Table 5). Therefore, we did not include tumor burden in the multivariate analysis in the initial submission. To satisfactorily address the reviewer's comment, we consulted Dr. J. Jack Lee, an experienced and well-respected biostatistician with expertise in clinical trials, biomarker and survival analysis, who has supervised all statistical analyses for this revision.

The typical steps for variable selection in the Cox model analysis are:

- a. Conduct univariate analysis on all variables.
- b. Select only the significant variables to be included in the initial Cox model.
- c. Perform an iterative analysis to drop the non-significant variables each time until all the remaining ones are significant.
- d. No adjustment of multiplicity is needed since the process of the stepwise Cox regression model has taken account of the joint effect of multiple variables.
- e. If one is interested in a particularly important variable, it can be included in the Cox model even though it is not significant. However, adding more non-significant variables will dilute the statistical power for identifying significant

variables.

Given the potential importance of tumor burden, we repeated multivariate analysis incorporating tumor burden as the reviewer suggested. After adjusting for tumor burden and stage, the association between ctDNA level and PFS or OS remained significant. We have updated Supplementary Table 5 and results (page 10, line 180-182, 190-191) accordingly, and added discussion (page 14, line 258-276) regarding to this important topic in the revised manuscript.

4. The authors identify 33 mutations that were seen only in post-treatment samples. The authors should more clearly define which of these are novel and not previously implicated in small cell lung cancer chemo resistance. The study would be strengthened by functional assays assessing whether these alterations contribute to chemo resistance.

Author's response: We thank the reviewer for this excellent suggestion and we agree that further discussion on mutations enriched in post-treatment samples will provide valuable information to readers. We did functional predication of all these mutations and added a supplementary table (Supplementary Table 7) and a paragraph to discuss about these 33 mutations (page 12, line 217-220), We also agree that functional analyses on the impact of mutations enriched in post-treatment samples on chemo resistance will be needed. However, considering the small sample size of this pilot study, confounding factors such as mutagenic effects of radiation and chemotherapy, we are planning to conduct functional assays on candidate genes/mutations after data from a large follow up cohort becomes available.

REVIEWERS' COMMENTS:

Reviewer #1 (Remarks to the Author):

The authors have addressed issues in the initial review and the manuscript is ready for publication.

Reviewer #2 (Remarks to the Author):

The authors have done a thorough job of addressing all of my concerns. In particular adding multivariate analysis including tumor burden was critical to the conclusions of their manuscript, which they have now done. I have no other concerns and think that this manuscript will be of high interest to the field.

We thank the reviewers for their efforts and we are glad that both reviewers were satisfied with our revised manuscript. We list the reviewers' comments as following.

Reviewer #1 (Remarks to the Author):

The authours have addressed issues in the initial review and the manuscript is ready for publication.

Reviewer #2 (Remarks to the Author):

The authors have done a thorough job of addressing all of my concerns. In particular adding multivariate analysis including tumor burden was critical to the conclusions of their manuscript, which they have now done. I have no other concerns and think that this manuscript will be of high interest to the field.